# *Toxocara cati* Infection in Cats (*Felis catus*): A Systematic Review and Meta-Analysis

**DOI:** 10.3390/ani14071022

**Published:** 2024-03-27

**Authors:** Jorge Luis Bonilla-Aldana, Alba Cristina Espinosa-Nuñez, D. Katterine Bonilla-Aldana, Alfonso J. Rodriguez-Morales

**Affiliations:** 1School of Veterinary Medicine and Zootehcnics, Faculty of Agricultural Sciences, Universidad de la Amazonia, Florencia 111321, Caquetá, Colombia; jorg.bonilla@udla.edu.co (J.L.B.-A.); a.espinosa@udla.edu.co (A.C.E.-N.); 2Research Unit, Universidad Continental, Huancayo 12001, Peru; 3Masters of Climate Change and Clinical Epidemiology and Biostatistics Program, Universidad Cientifica del Sur, Lima 15307, Peru; arodriguezmo@cientifica.edu.pe; 4Gilbert and Rose-Marie Chagoury School of Medicine, Lebanese American University, Beirut P.O. Box 36-5053, Lebanon

**Keywords:** *Toxocara*, prevalence, cats, systematic review, meta-analysis

## Abstract

**Simple Summary:**

Toxocariasis, caused by species of *Toxocara*, affects canines, felines, humans, and other vertebrates. The primary mode of infection is by ingesting embryonated eggs. It poses environmental, human, and animal health risks, especially in park soils. This study aimed to assess the global prevalence of *Toxocara cati* in cats (*Felis catus*), a neglected species compared to *T. canis*, via a systematic literature review across six databases. Significant prevalence was observed using coproparasitological methods, with Nepal displaying the highest rates. The findings highlight the imperative of preventive measures against toxocariasis due to its widespread occurrence. Recognising the interconnectedness of animal, environmental, and human health underscores the importance of deworming cats, promoting hygiene, and educating the public to mitigate the risks of this zoonotic condition. Protecting feline health benefits cats and reduces the likelihood of human transmission, creating a positive outcome for both.

**Abstract:**

**Introduction:** Toxocariasis is an infection caused in canines, felines, humans, and other vertebrates by species of the genus *Toxocara*, such as *T. canis* and *T. cati*. The embryonated eggs of these parasites are the primary means of acquiring the infection for both definitive hosts, dogs and cats, respectively, and for intermediates, such as humans and other vertebrates. When deposited on park soils, environmental contamination becomes a risk to environmental, human, and animal health. **Objective:** To determine the global prevalence of *Toxocara cati* in cats (*Felis catus*). **Methods:** A systematic review of the literature was carried out in six databases (Scopus, PubMed, ScienceDirect, SciELO and Google Scholar) to evaluate the global prevalence of *Toxocara cati* in cats, defined by coproparasitological, histological, and molecular techniques. A meta-analysis was performed using a random effects model to calculate pooled prevalence and 95% confidence intervals (95% CI). A two-tailed 5% alpha level was used for hypothesis testing. **Results:** Two hundred and eighty-nine studies were included. The global pooled prevalence of *Toxocara cati* in cats using coproparasitological methods was 17.0% (95.0% CI: 16.2–17.8%). In the subgroup analysis according to country, Nepal had the highest prevalence of *T. cati* infection (94.4%; 95% CI 89.7–99.2%). The pooled prevalence of *T. cati* infection by PCR in four studies was 4.9% (95.0% CI: 1.9–7.9%). **Conclusions:** This systematic review underscores the need for preventive action against toxocariasis due to its widespread prevalence. The interplay between animal and human health should be emphasised, necessitating measures like deworming cats, hygiene practices, and public education to mitigate risks. Safeguarding feline health can also reduce human transmission, benefiting both species.

## 1. Introduction

Zoonoses are a group of infectious diseases transmissible between animals and humans [1], including conditions where the human is not a definitive host of the etiological agent [2]. Pets such as dogs and cats are considered opportune hosts of various pathogenic agents of zoonotic incidence, such as gastrointestinal helminths of the *Toxocara* genus [3]. Toxocariasis is a parasitic disease with worldwide distribution, and its etiological agents in dogs and cats are *Toxocara canis* and *Toxocara cati*, respectively [4]. The parasite is transmitted vertically (transplacental and transmammary) and horizontally, addressing the ingestion of embryonated eggs in infected animals’ soil and fur and through consuming contaminated food [5,6,7,8]. The adult nematodes of *T. canis* and *T. cati* complete their reproductive cycle in the intestine of their definitive host (dogs and cats), reproducing and eliminating about 200,000 eggs per day, excreted in faeces into the environment [9]. 

In dogs and cats, *T. canis* and *T. cati* mainly affect young animals from birth, presenting clinical signs such as cachexia, emaciation, body weakness, rough coat, growth dekay, vomiting, cough, diarrhoea, and distended abdomen; the cough is due to larval migration to the lungs [10,11,12]. The disease can affect adult cats and dogs, but they do not usually present clinical signs [13]. *Toxocara* infection occurs accidentally in humans due to the ingestion of eggs in soil or contaminated food, including paratenic hosts, such as poultry, pork, and beef [14,15]. Based on the clinical manifestations observed in humans, the disease can be classified into four primary syndromes: visceral larva migrans (VLM), ocular toxocariasis (OT), covert toxocariasis (TC), and neuro toxocariasis (NT) [16]. 

At a global level, both stray and domestic cats contribute to the dispersion and contamination of embryonated *Toxocara* eggs into the environment [17,18]. The presence of eggs in public places represents a risk for animal health and even for humans, given that approximately 21% of public spaces worldwide are contaminated with *Toxocara* eggs [19]. In Latin America, it is estimated that the prevalence of *Toxocara* in public parks is 50%, which means it can be considered a transmission route for people who attend these places, mainly children who might play on the ground [20]. On the other hand, studies have reported that direct contact with the fur of cats infected with *T. cati* is a route of transmission since potentially infective embryonated eggs have been identified in perianal areas, extremities, and the lower part of the tail of cats [8,21]. The global prevalence of *T. cati* in cats starts from 17%, with an average of 134 million cats worldwide contributing to the dispersal of eggs in the environment, generating a public health problem [11]. 

Diagnostic tests for *Toxocara* spp. in pets are fundamentally based on the microscopic examination of faeces to find eggs and analyse their morphology, using different coprodiagnostic techniques such as direct smear, Kato–Katz, MacMaster, and Faust (flotation–sedimentation), among others [22]. However, other diagnostic tests have greater sensitivity and specificity, such as serological tests, ELISA to detect anti-*Toxocara* IgG antibodies, and molecular techniques, such as polymerase chain reaction (PCR) and Western blot, that determine the larval TES antigen [23,24,25].

## 2. Methods

**Protocol:** The protocol followed the recommendations established by the PRISMA statement.

**Inclusion criteria**: Peer-reviewed published articles were included in which infection with coproparasitological, histological, or molecular confirmation of *Toxocara catis* in cats (*Felis catus*) was reported. For parasitological tests, we considered egg detection for tests based on molecular biology and PCR. The article language limit was not set, and we included publications from 1 January 1950 to the date the search ended, 31 January 2024. Review articles, opinion articles, and letters that do not present original data were excluded from the study, as were studies that reported cases with incomplete information.

**Information sources and search strategy**: A systematic review was conducted using Medline/PubMed, Scopus, ScienceDirect, SciELO, and Google Scholar. The search terms used were the following: “Prevalence”, “*Toxocara*”, “*Toxocara cati*”, and “cats”.

**Study selection**: Results from the initial search strategy were first selected by title and abstract. The full texts of relevant articles were examined for the inclusion and exclusion criteria. When an article provided duplicate information from the same subjects, the information from both reports was combined to obtain complementary data, counting as only one study. Observational studies reporting the prevalence of *Toxocara cati* in cats were included for quantitative synthesis (meta-analysis).

**Data collection process and data elements**: Two researchers independently completed data extraction forms, including information on publication type, publishing institution, country, year and date of publication, and number of infected animals evaluated by serological or molecular tests. A third researcher verified the list of articles and data extractions to ensure no duplicate articles or information from the same study were presented, and resolved any discrepancies regarding study inclusion.

**Assessment of methodological quality and risk of bias**: We used the IHE case series study quality assessment checklist and the critical appraisal tool to assess the quality of cross-sectional studies (AXIS) [26]. Publication bias was assessed using a funnel plot. Given the varying degrees of data heterogeneity and the heterogeneity inherent in any systematic review of published literature studies, a random effects model was used to calculate the pooled prevalence and 95% confidence interval (95%CI). 

**Statistical approach**: Unit discordance for variables was resolved by converting all units to a standard measurement for that variable. Percentages and means ± standard deviation (SD) were calculated to describe the distributions of categorical and continuous variables, respectively. Since individual case information was unavailable, we will report weighted means and SDs. Baseline data were analysed using Stata version 14.0, licensed. Meta-analyses were performed with Stata, the licensed Open Meta [Analyst], and Comprehensive Meta-Analysis ve.3.3^®^ software. The pooled prevalences and their 95% confidence intervals (95% CI) were used to summarise the weighted effect size for each study pooling variable using the binary random effects model of the individual studies (weighting took into account sample sizes), except for median age, where a continuous random effects model was applied (DerSimonian–Laird procedure). A random effects meta-analysis model will imply the assumption that the effects estimated in the different studies are not identical but rather follow a particular distribution. For random effects analyses, the pooled estimate and 95% CIs refer to the centre of the pooled prevalence distribution but do not describe the width of the distribution. Often, the pooled estimate and its 95% CI are cited in isolation as an alternative estimate of the quantity evaluated in a fixed-effects meta-analysis, which is inappropriate. The 95% CI of a random effects meta-analysis describes the uncertainty in the location of the systematically different mean prevalence in different studies. Measures of heterogeneity, including Cochran’s Q statistic, I^2^ index, and squared tau test, were estimated and reported. We performed subgroup analyses using techniques, countries, subregions, and meta-analyses for each variable of interest. Publication bias was assessed using a funnel plot. A random effects model was used to calculate pooled prevalence and 95% CI, given the varying degrees of data heterogeneity and inherent heterogeneity in any systematic review of published literature studies.

## 3. Results

### 3.1. Selection of Studies

Our search strategy yielded 16,266 records in the databases combined. After removing duplicates and screening for titles and abstracts, 329 articles underwent full-text review. Finally, 289 articles were included in the systemic review and meta-analysis [27,28,29,30,31,32,33,34,35,36,37,38,39,40,41,42,43,44,45,46,47,48,49,50,51,52,53,54,55,56,57,58,59,60,61,62,63,64,65,66,67,68,69,70,71,72,73,74,75,76,77,78,79,80,81,82,83,84,85,86,87,88,89,90,91,92,93,94,95,96,97,98,99,100,101,102,103,104,105,106,107,108,109,110,111,112,113,114,115,116,117,118,119,120,121,122,123,124,125,126,127,128,129,130,131,132,133,134,135,136,137,138,139,140,141,142,143,144,145,146,147,148,149,150,151,152,153,154,155,156,157,158,159,160,161,162,163,164,165,166,167,168,169,170,171,172,173,174,175,176,177,178,179,180,181,182,183,184,185,186,187,188,189,190,191,192,193,194,195,196,197,198,199,200,201,202,203,204,205,206,207,208,209,210,211,212,213,214,215,216,217,218,219,220,221,222,223,224,225,226,227,228,229,230,231,232,233,234,235,236,237,238,239,240,241,242,243,244,245,246,247,248,249,250,251,252,253,254,255,256,257,258,259,260,261,262,263,264,265,266,267,268,269,270,271,272,273,274,275,276,277,278,279,280,281,282,283,284,285,286,287,288,289,290,291,292,293,294,295,296,297,298,299,300,301,302,303,304,305,306,307,308,309,310] (Table 1). Figure 1 shows the PRISMA flow chart.

### 3.2. Characteristics of Included Studies 

The characteristics of the included articles are summarised in Table 1. A total of 289 articles were included, in which 168,643 cats were evaluated, 92.6% by coproparasitological techniques, 5.4% by necropsy (histology), and 2.0% by PCR. The studies ranged from 1973 to 2023, but there were 30 (7.71%) in 2019 (Table 1). The studies were distributed as follows: Brazil (71 studies), the United States (23 studies), Italy (22 studies), Iran (21 studies), and Portugal (19 studies), among other 57 countries (Table 1). All faecal samples were evaluated using Direct Smear, Graham, Kinyou, Mini Parasep^®^Sf, Sporulation, Flotac, Bailinger, Mcmaster, Wisconsin, Centrifugal Flotation, Acid-Fast, Flotation (Faust), Charles, Meriflour, Mini-Flotac, Concentration Flotation, Sedimentation (Hoffman), Centrifugal Sedimentation, Baermann, Mifc (Merthiolate-Iodine-Formaldehyde-Concentration), Modified Telemann, Teuscher, Fulleborn, Fecal Smear (Ziehl-Neelsen), Sheater, Formalin Ether (Ritchie), Willis, and Gordon E. Whitlock techniques, among others, searching for *Toxocara* eggs, larvae, and adult parasites. PCR was also used from faecal samples to detect *Toxocara cati*. 

### 3.3. Risk of Bias Assessment

In the risk of bias assessment, twenty studies were at high risk of bias, while the remaining 269 were at low risk of bias.

### 3.4. Prevalence of Toxocara cati in Cats Found Using Coproparasitological Methods

The global pooled prevalence of *Toxocara cati* in cats found using coproparasitological methods was 17.0% (95.0% CI: 16.2–17.8%), with high heterogeneity (I^2^ = 97.981%, τ^2^ = 0.004, Q^2^ = 15652.788) (Figure 2). In the subgroup analysis by year (Figure 3), 1996 was the year with the highest reported pooled prevalence of *Toxocara cati* infection (90.6%; 95% CI 73.9–100.0%), followed by 1991 (60.0%; 95% CI 42.5–77.5%), and 2001 (36.2%; 95% CI 19.2–53.2%) (Figure 3). In the subgroup analysis according to country (Figure 4), Nepal had the highest prevalence of *Toxocara cati* infection (94.4%; 95% CI 89.7–99.2%), followed by the United Kingdom (90.9%; 95% CI 73.9–100.0%) and Bangladesh (76.9%; 95% CI 54.0–99.8%), among other countries (Figure 4). In the subgroup analysis according to continents or regions (Figure 5), Asia had the highest prevalence of *Toxocara cati* infection (27.9%; 95% CI 24.5–31.4%) (I^2^ = 99.11%), followed by Africa (21.4%; 95% CI 7.1–35.6%) (I^2^ = 93.03%) and North America (18.5%; 95% CI 15.2–21.9%) (I^2^ = 98.1%) (Figure 5).

Considering the types of cats (Feral, Stray, Shelter, Domestic, and Breed), we found that the highest prevalence of *Toxocara cati* infection was in feral cats (42.6%; 95% CI 29.8–55.4%) (I^2^ = 96.72%), followed by stray cats (29.9%; 95% CI 25.3–34.4%) (I^2^ = 98.34%) and shelter cats (20.1%; 95% CI 16.1–24.1%) (I^2^ = 97.79%) (Figure 6).

Regarding the coproparasitological methods, we found that the highest prevalence of *Toxocara cati* infection was obtained through a faecal direct smear (26.1%; 95% CI 22.7–29.5%) (I^2^ = 98.77%), followed by flotation (Faust) (19.9%; 95% CI 18.4–21.4%) (I^2^ = 98.46%) and centrifugal flotation (16.2%; 95% CI 12.3–20.1%) (I^2^ = 96.71%) (Figure 7).

### 3.5. Prevalence of Toxocara Found Using Necropsy (Histology)

The pooled prevalence of toxocariasis found using the necropsy (histology) of gastrointestinal tissues was 30.0% (95.0% CI: 26.1–33.8%) with high heterogeneity (I^2^ = 98.64%, τ^2^ = 0.022, Q^2^ = 4545.122) (Figure 8).

### 3.6. Prevalence of Toxocara catis Found Using PCR

The pooled prevalence of *Toxocara cati* infection found using PCR in four studies (N = 3454) was 4.9% (95.0% CI: 1.9–7.9%) with high heterogeneity (I^2^ = 97.48%, τ^2^ = 0.001, Q^2^ = 119.225) (Figure 9).

## 4. Discussion

Toxocariasis, a helminth parasitic disease, is widespread, particularly in low- and middle-income nations. Despite its significant clinical implications, including the potential for fatal outcomes in humans and animals, mainly domestic ones, such as dogs and cats, many countries, particularly those with limited resources, do not actively monitor this condition [312]. There is a lack of epidemiological surveillance in many regions of the planet for toxocariasis in humans and animals.

This systematic review and meta-analysis, aimed at determining the pooled prevalence of *Toxocara cati* in cat populations worldwide using a comprehensive search approach across six databases, found a relevant prevalence. The findings underscore the considerable diversity in parasite prevalence across various countries and continents, as indicated by previous studies [20,313,314]. Through an extensive exploration of studies published between 1973 and 2023 across diverse geographic regions, this study facilitated the execution of a meta-analysis to ascertain the global prevalence of *T. cati*. This broad temporal and geographical scope enabled a robust synthesis of data, based on more than 150,000 animals, to provide insights into the prevalence patterns of this parasite on a global scale. As expected, *Toxocara cati*, compared with *T. canis*, is more neglected [315], making it difficult to understand that this pathogen affects other domestic and non-domestic animals and humans [316,317,318,319]. Toxocariasis in humans is also neglected, especially in developing countries [320,321]. Very few studies, and even more cases reported, can confirm *T. cati* infection specifically in humans, by serological or molecular methods, as compared with just toxocariasis or *Toxocara* spp infection in humans, mainly due to a lack of confirmation or a lack of specific tests to confirm species at diagnosis. It is also believed that there are no implications at all regarding the implicated *Toxocara* species [316,322]. Recent studies suggest that no proteins from *T. canis* and *T. cati* exist that could be used as a diagnostic tool to enable differential serodiagnostics of these species in humans. In addition, a heterogenic protein pattern between individual hosts has been found, which was most pronounced in *T. cati*-infected pigs [322]. 

Most of the studies on toxocariasis in animals have traditionally focused on dogs and *Toxocara canis* [12,323,324]. Comparatively, there have been a lack of studies on toxocariasis in cats, mainly due to *T. cati*. Cats are also relevant hosts of zoonotic diseases, specifically zoonotic parasites [325,326]. In this systematic review, it was observed that coproparasitological methods are still the predominant means of establishing toxocariasis in cats, showing a relevant prevalence that seems to be higher during specific years and places, probably, as shown before, influenced by seasonal, environmental, and even climatic factors [327,328,329,330]. For example, as expected, the country with the highest prevalence was Nepal, a country with a low Human Capital Index (0.5) and included in the group of lower-middle income economies (LMIE), according to the World Bank (https://datahelpdesk.worldbank.org/knowledgebase/articles/906519-world-bank-country-and-lending-groups) (accessed on 1 February 2024). Higher prevalences were also observed in other LMIE, such as Bangladesh, Vietnam, and Myanmar, all of them in Asia, which resulted in it being the continent with the highest prevalence. Culturally, there is a high level of contact with and apparent care of cats by humans in many Asian and Middle Eastern countries, such as Turkey (17%), Egypt (30%), and China (11%), among others. A recent study showed that cats are more popular than dogs in 91 countries, and dogs are more prevalent in 76 countries (https://www.budgetdirect.com.au/pet-insurance/guides/cats-vs-dogs-which-does-the-world-prefer.html) (accessed on 1 February 2024). However, the number of articles and the number of samples analysed per study for some countries would be insufficient to understand the relationships between prevalence and associated factors, despite the fact that the prevalence is weighted in the meta-analysis by the number of studies and the sample size. 

The type of cat significantly influences the prevalence of toxocariasis in cats. Those living in wild, non-urban areas (feral) presented the highest prevalence (43%), while domestic cats (13%) and breed cats (3%) showed the lowest values. Other studies show that this is a risk factor for higher prevalences [326]. In general, unattended cats without proper veterinary control and assessment are at risk of exposure and infection.

The main coproparasitological methods vary slightly regarding the prevalence of toxocariasis, from 10.5% to 26.1%, with the faecal direct smear method associated with the highest prevalence.

Many studies assessed infection in dead animals, reporting a high prevalence, even higher than those studies assessing infection by coproparasitological methods. The prevalence of *Toxocara cati* at necropsy was 30%. In contrast, PCR prevalence was only 5%. Then, this was less sensitive than coproparasitological methods (17%). Again, the number of articles and the number of samples analysed per study by molecular methods such as PCR, would be insufficient to understand the differences in the sensitivity and specificity of methods, despite the fact that the prevalence is weighted in the meta-analysis by the number of studies and the sample size. To understand the sensitivity of PCR, specific studies of diagnostic test comparison should be performed, which was clearly outside the objectives of this systematic review, which focused on the prevalence of *T. cati* in cats. 

Indeed, the histological diagnosis of *T. cati* can be limited, which makes a differential diagnosis with *T. canis* impossible. An infection due to *Toxocara* in a cat is not necessarily due to *T. cati*, as an infection due to *Toxocara* in a dog is not necessarily due to *T. canis*. Both species may infect other hosts, and in some, these may serve as paratenic hosts, just serving for infection without the reproduction and development of adult forms, as occurs in humans that are exclusively paratenic hosts [331,332,333]. More commercial tests and laboratories with standardised PCR for molecular diagnosis must be conducted. The molecular diagnosis of toxocariasis is only sometimes available for humans, where the primary tool is serological tests [312,333,334]. There is an urgent need for a molecular diagnosis of toxocariasis, with possibilities of sequencing and identifying species [335,336]. At the same time, better immunological tests are required, as ELISA and Western blot still need to be improved, mainly due to the antigen quality. Then, recombinant antigens-based tests are preferred and it is recommended that they are widely available [337,338,339].

Although dogs have been studied more, the present systematic review shows that infections due to *Toxocara cati* in cats may be even higher (17%) than those due to *T. canis* in dogs. A recent systematic review of *T. canis* in dogs found that the overall prevalence was 11.1% (95% CI, 10.6–11.7%) after studying more than 3 million dogs in 60 countries [12]. The authors concluded that young (<1 year of age), stray, rural, and male dogs had a significantly higher prevalence of infection than older, pet, urban, or female dogs [12]. Our results confirm the findings of a review from 2020, in which the prevalence of *Toxocara* infection in cats was 17.0% (16.1–17.8%), but there was a contrast regarding the continents, as this review found the highest prevalence in African countries (43.3%, 28.3–58). As mentioned earlier, we found this in Asia (28%). In Africa, we found 21.4 (7.1–35.6%). They found that the prevalence of *Toxocara* was higher in stray cats (28.6%, 25.1–32.1%) [11]. Our review found 29.9% in stray cats (25.3–34.4%) but higher results in feral cats (42.6%). 

As indicated, cats may also serve as a source of human infections due to *Toxocara*. Cats play a crucial role globally as primary hosts for *Toxocara*, releasing eggs into the environment and thereby heightening public health concerns. Health authorities and cat caregivers must prioritise efforts toward preventing and managing this zoonotic disease in feline populations. This is especially crucial in regions with elevated risk factors and prevalence rates, necessitating heightened vigilance and proactive measures [11].

Cats, as with other species, may also be infected with another member of the family *Toxocaridae*, as is the case of *Toxascaris leonina*; nevertheless, fewer studies about it are available [340,341]. Regardless, studies and systematic reviews so far are lacking and needed [341].

This systematic review has certain limitations, including the fact that we were unable to assess the age or gender of cats, as this was not reported in most of the studies. This aspect could also be important in the prevalence and risk of *T. cati* infection, as has been suggested in *T. canis* [312].

## 5. Conclusions

The significance of toxocariasis in cats is its potential to infect humans, in addition to the damage that it may cause to felines. Humans can become accidental hosts by ingesting *Toxocara cati* eggs through contaminated soil, water, or food. Once ingested, the larvae can migrate to various tissues in the body, causing visceral larva migrans (VLM) or ocular larva migrans (OLM), which can result in serious health complications, including vision impairment, organ damage, and even neurological disorders. Preventive measures should be considered, given the high prevalence found in this systematic review and previous studies. The zoonotic aspect of toxocariasis in cats and dogs highlights the interconnectedness of animal and human health, including OneHealth, emphasising the importance of preventive measures such as deworming protocols for cats [311,342,343,344], proper hygiene practices, and public education on the risks associated with exposure to contaminated environments. By addressing toxocariasis in cats, feline health can be safeguarded, and the potential transmission of this parasitic infection to humans and other animals can also be minimised, promoting the well-being of both animals and humans. Finally, in farming animals, toxocariasis can lead to reduced productivity. Infected animals may exhibit decreased weight gain, reduced milk production (in dairy cattle), decreased fertility, and lower overall performance. This can directly impact farm income by reducing the quantity and quality of products produced.

## Figures and Tables

**Figure 1 animals-14-01022-f001:**
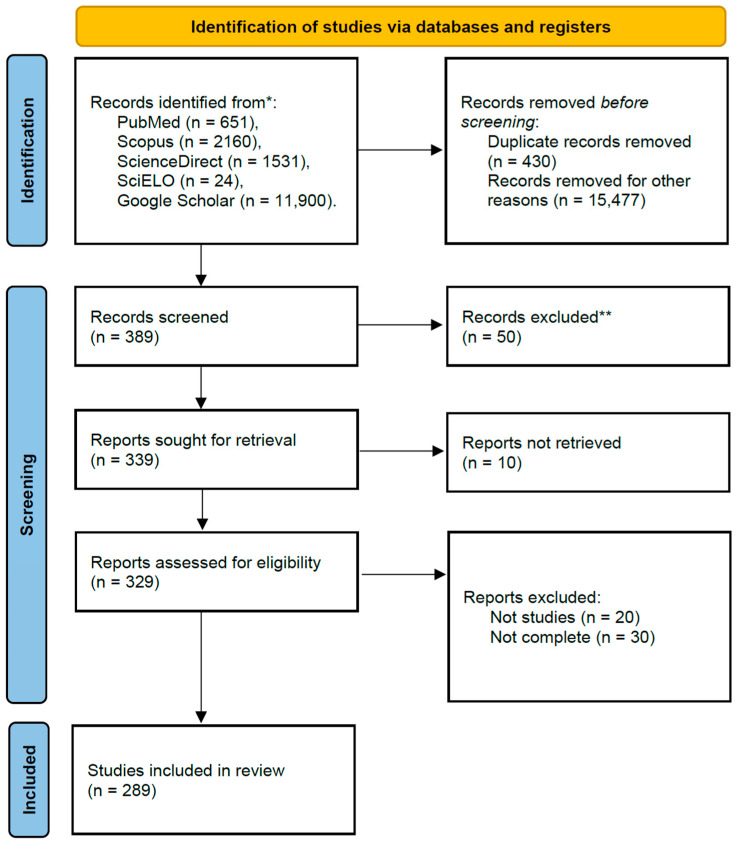
The 2020 PRISMA flow diagram. * All included databases, raw results. ** At an initial quality screening, including lack of inclusion criteria.

**Figure 2 animals-14-01022-f002:**
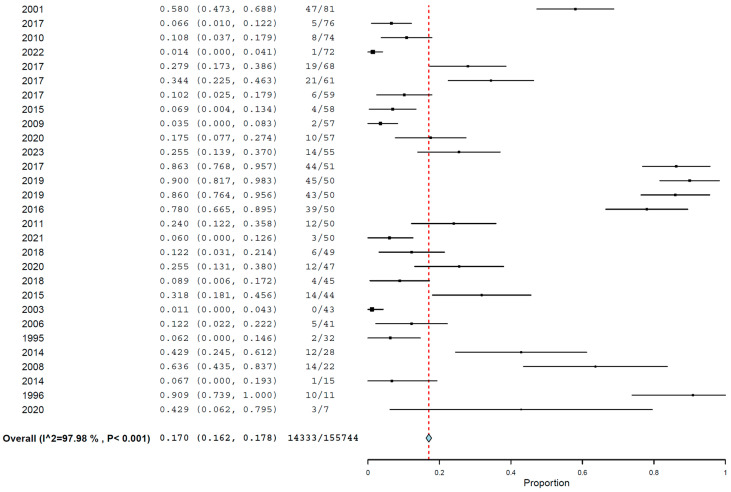
Prevalence of *Toxocara cati* in cats found using coproparasitological methods.

**Figure 3 animals-14-01022-f003:**
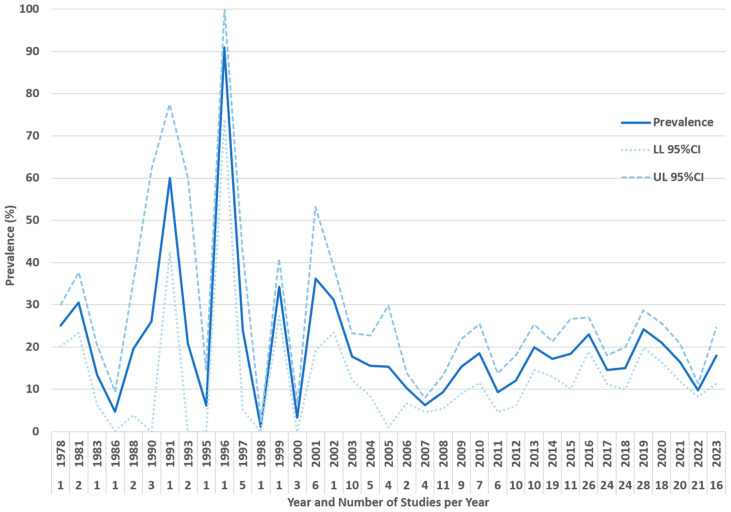
Prevalence of *Toxocara cati* in cats found using coproparasitological methods by years.

**Figure 4 animals-14-01022-f004:**
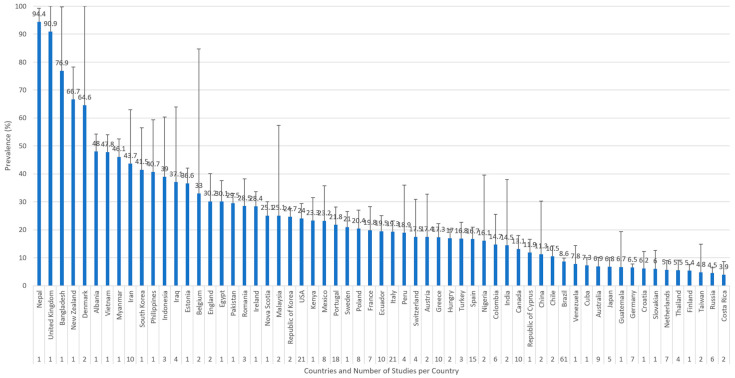
Prevalence of *Toxocara cati* in cats found using coproparasitological methods by countries. Error bars show the upper 95% CI value.

**Figure 5 animals-14-01022-f005:**
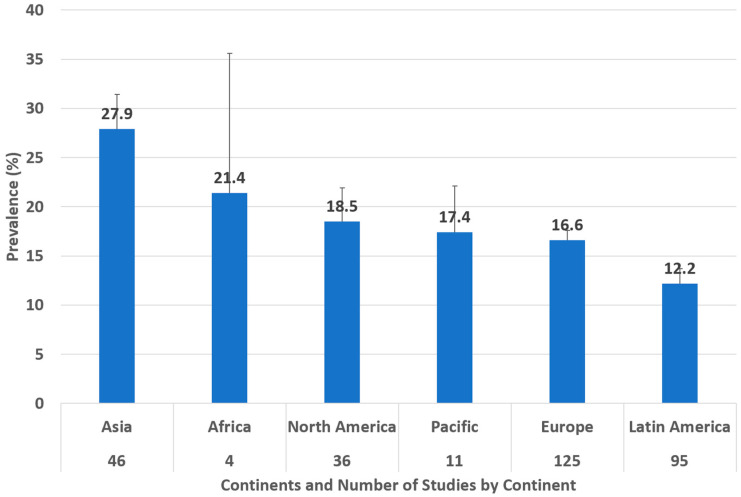
Prevalence of *Toxocara cati* in cats found using coproparasitological methods by continents. Error bars show the upper 95% CI value.

**Figure 6 animals-14-01022-f006:**
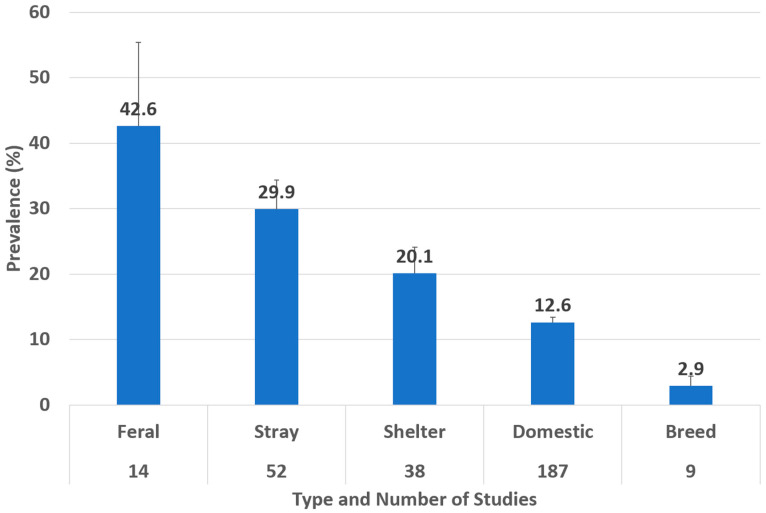
Prevalence of *Toxocara cati* in cats found using coproparasitological methods by cat type. Error bars show the upper 95% CI value.

**Figure 7 animals-14-01022-f007:**
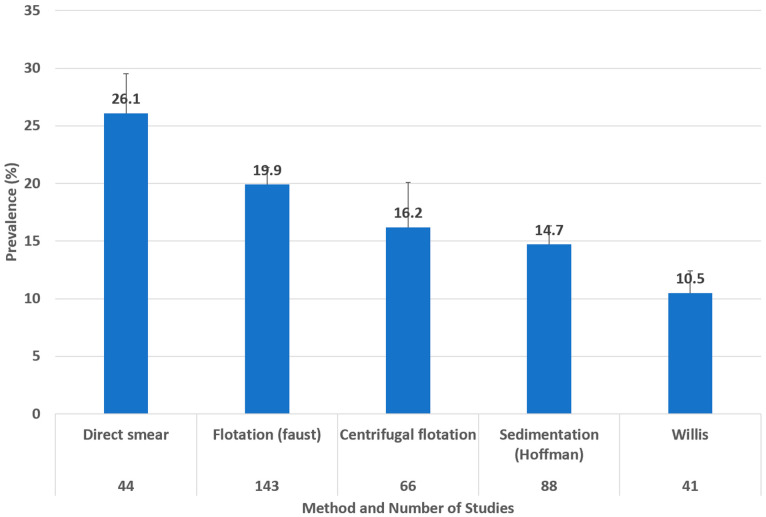
Prevalence of *Toxocara cati* in cats found using different coproparasitological methods. Error bars show the upper 95% CI value.

**Figure 8 animals-14-01022-f008:**
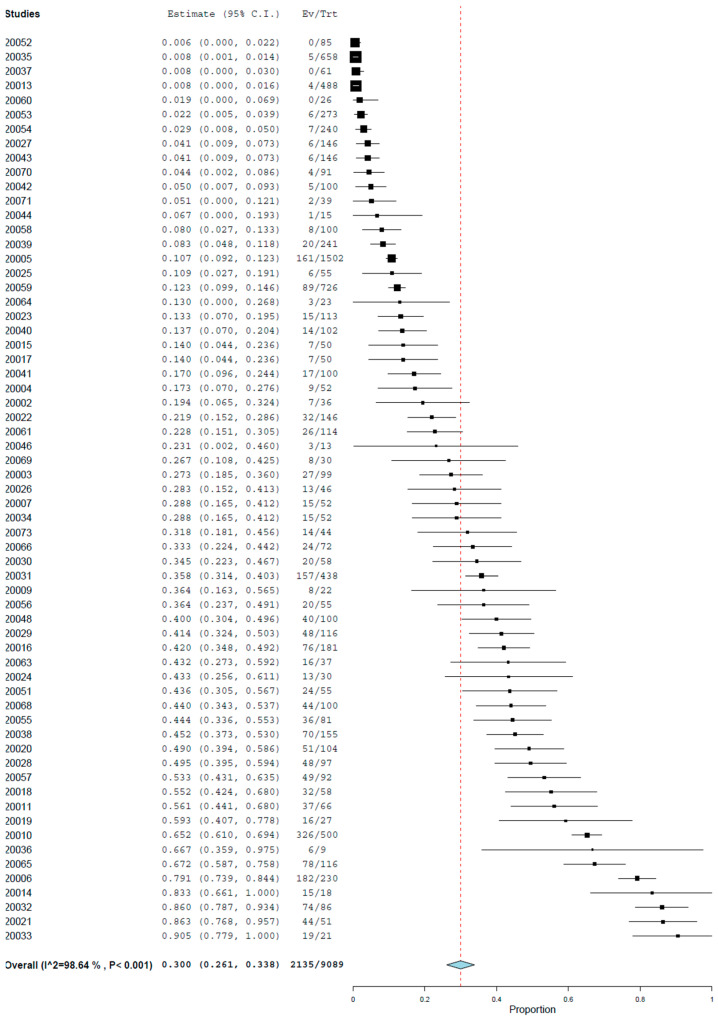
Prevalence of toxocariasis in cats found using necropsy (histology).

**Figure 9 animals-14-01022-f009:**
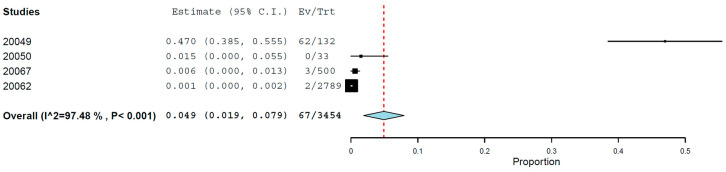
Prevalence of *Toxocara cati* in cats found using PCR.

**Table 1 animals-14-01022-t001:** Studies included.

Code	Study Title	Publication	Study	Location	Country	N	Ref.
RSM-1	Prevalence Of Intestinal Canine And Feline Parasites In Saitama Prefecture, Japan	2009	1999	Saitama	Japan	1079	[27]
RSM-2	Prevalencia De Helmintos Intestinales En Gatos Domésticos Del Departamento Del Quindío, Colombia	2012	2008	Quindío	Colombia	121	[28]
RSM-3	10-Year Parasitological Examination Results (2003 To 2012) Of Faecal Samples From Horses, Ruminants, Pigs, Dogs, Cats, Rabbits And Hedgehogs	2017	2003		Germany	903	[29]
RSM-4	A Comparative Study Of Some Intestinal Parasites In Fecal Samples Of Domestic And Stray Cats In Baghdad, Iraq	2022	2020	Baghdad	Iraq	121	[30]
RSM-5	A Cross-Sectional Study Of Tritrichomonas Foetus Infection In Feral And Shelter Cats In Prince Edward Island, Canada	2016	2011	Prince Edward Island	Canada	100	[31]
RSM-6	A Retrospective Investigation Of Feline Gastrointestinal Parasites In Western Canada	2013	1998		Canada	635	[32]
RSM-7	A Survey Of Gastrointestinal Helminths In Cats Of The Metropolitan Region Of Rio De Janeiro, Brazil	2004	2004	Rio de Janeiro	Brazil	135	[33]
RSM-8	A Survey Of Helminth Parasites Of Cats From Saskatoon	1999	1999	Saskatoon	Canada	52	[34]
RSM-9	A Survey Of Helminths In Domestic Cats In The Pretoria Area Of Transvaal, Republic Of South Africa, Part 1: The Prevalence And Comparison Of Burdens Of Helminths In Adult And Juvenile Cats	1989	1980		South Africa	1502	[35]
RSM-10	A Survey Of Helminths In Stray Cats From Copenhagen With Ecological Aspects	1984	1980	Copenhagen	Denmark	230	[36]
RSM-11	A Survey Of *Toxocara* Infections In Cat Breeding Colonies In The Netherlands	1998	1995		Netherlands	337	[37]
RSM-12	A Survey On Endoparasites And Ectoparasites In Domestic Dogs And Cats In Vladivostok, Russia 2014	2016	2013	Vladivostok,	Russia	54	[38]
RSM-13	A Survey On Endoparasites And Ectoparasites Of Stray Cats From Mashhad (Iran) And Association With Risk Factors	2011	2009	Mashhad	Iran	52	[39]
RSM-14	A Survey On The Prevalence Of *Toxocara cati*, *Toxocara* Canis And Toxascaris Leonina Eggs In Stray Dogs And Cats’ Faeces In Northwest Of Iran: A Potential Risk For Human Health	2019	2017	Azarshahr, Marand	Iran	100	[40]
RSM-15	A Survey On *Toxocara cati* Eggs On The Hair Of Stray Cats: A Potential Risk Factor For Human Toxocariasis In Northeastern Iran	2019	2016	Mashhad	Iran	167	[8]
RSM-16	A Survey Study On Gastrointestinal Parasites Of Stray Cats In Azarshahr, (East Azerbaijan Province, Iran)	2016	2013	Azarshahr	Iran	50	[41]
RSM-17	Abundance, Zoonotic Potential And Risk Factors Of Intestinal Parasitism Amongst Dog And Cat Populations: The Scenario Of Crete, Greece	2017	2011	Crete	Greece	264	[42]
RSM-18	An Investigation Of The Potential For Spread Of *Sarcocystis* spp. And Other Parasites By Feral Cats	1990	1984		New Zealand	63	[43]
RSM-19	Aporte Al Conocimiento De Los Metazoos Parásitos Del Gato Doméstico En El Departamento De Montevideo, Uruguay.	2013	2002	Montevideo	Uruguay	22	[44]
RSM-20	Avaliação Das Endoparasitoses Intestinais Que Acometem Cães E Gatos Mantidos Em Um Abrigo	2021	2019	Zona da Mata Mineira	Brazil	26	[45]
RSM-21	Frequência De Parasitoses Com Potencial Zoonótico Em Cães E Gatos Naturalmente Infectados Na Cidade De Maringá-PR	2022	2021	Maringá	Brazil	19	[46]
RSM-22	Canine And Feline Helminth And Protozoan Infections In Belgium	1973	1973		Belgium	500	[47]
RSM-23	Caracterização Da Ocorrência De Parasitas Gastrointestinais De Gatos Na Zona De Pesca Da Ilha De Faro	2018	2014	Faro	Portugal	123	[48]
RSM-24	Caracterização Molecular De *Cryptosporidium* Spp. E Ocorrência Dos Principais Parasitas Gastrointestinais Em Amostras Fecais De Cães E Gatos Naturalmente Infectados	2018	2017		Brazil	49	[49]
RSM-25	Caracterización De La Infestación Parasitológica Gastrointestinal Y Respiratoria En Gatos Ferales (Felis Silvestris Catus) De La Ciudad De Córdoba, Comunidad Autónoma De Andalucía, España.	2022	2022	Cordova	Spain	33	[50]
RSM-26	Cardiorespiratory Nematodes And Co-Infections With Gastrointestinal Parasites In New Arrivals At Dog And Cat Shelters In North-Western Spain	2022	2019	Galicia	Spain	65	[51]
RSM-27	Challenging The Dogma Of The ‘Island Syndrome’: A Study Of Helminth Parasites Of Feral Cats And Black Rats On Christmas Island	2018	2018	Christmas Island	Australia	66	[52]
RSM-28	Characterisation Of Ecto- And Endoparasites In Domestic Cats From Tirana, Albania	2014	2008	Tirana	Albania	252	[53]
RSM-29	Co-Infection Of Intestinal Helminths In Humans And Animals In The Philippines	2022	2019		Philippines	27	[54]
RSM-30	Comparação Da Prevalência De Parasitos Entéricos Em Gatos Errantes E Domiciliados Em Goiânia-Goiás, Análise Da Acurácia De Técnicas Parasitológicas E Avaliação Da Copro-Pcr Para O Diagnóstico De Toxoplasma Gondii	2016	2015	Goiânia	Brazil	149	[55]
RSM-31	Comparative Clinical Epidemiology Of Toxocariosis In Dogs And Cats	2010	2007	Lahore	Pakistan	671	[56]
RSM-32	Comparison Of *Toxocara* Eggs In Hair And Faecal Samples From Owned Dogs And Cats Collected In Ankara, Turkey	2014	2014	Ankara	Turkey	100	[21]
RSM-33	Coprological Detection Of Toxocariosis In Domicile And Stray Dogs And Cats In Sulaimani Province, Iraq	2022	2020	Sulaymaniyah	Iraq	78	[57]
RSM-34	Cross-Sectional Survey Of Toxoplasma Gondii Infection In Colony Cats From Urban Florence (Italy)	2010	2010	Florence	Italy	50	[58]
RSM-35	Cross-Sectional Survey On Tritrichomonas Foetus Infection In Italian Cats	2016	2012		Italy	267	[59]
RSM-36	*Cryptosporidium* Spp. And Other Zoonotic Enteric Parasites In A Sample Of Domestic Dogs And Cats In The Niagara Region Of Ontario	2006	2001	Ontario	Canada	41	[60]
RSM-37	Current Status Of L. Infantum Infection In Stray Cats In The Madrid Region (Spain): Implications For The Recent Outbreak Of Human Leishmaniosis?	2014	2014	Madrid	Spain	287	[61]
RSM-38	Descriptive Epidemiology Of Intestinal Helminth Parasites From Stray Cat Populations In Qatar	2008	2006	Doha	Qatar	488	[62]
RSM-39	Detection Of Helminth Eggs And Identification Of Hookworm Species In Stray Cats, Dogs And Soil From Klang Valley, Malaysia	2015	2013	Klang Valley	Malaysia	152	[63]
RSM-40	Determinación De La Presencia De Enteroparásitos En Gatos Clínicamente Sanos En Cuatro Comunas De Santiago, Mediante Los Métodos De Teuscher Y Teleman	2018	2018	Santiago de Chile	Chile	40	[64]
RSM-41	Determinación De La Presencia De Helmintos Gastro Intestinales En Gatos En Las Parroquias Urbano Marginales De La Ciudad De Babahoyo	2023	2023	Babahoyo	Ecuador	60	[65]
RSM-42	Determinación De Prevalencia De Parásitos Intestinales Y Externos En Gatos Domésticos (Felis Catus) En Determinadas Zonas Del Ecuador	2012	2012	Quito, Manta	Ecuador	40	[66]
RSM-43	Diagnosis Of Feline Whipworm Infection Using A Coproantigen ELISA And The Prevalence In Feral Cats In Southern Florida	2018	2014	Miami	USA	35	[67]
RSM-44	Diagnóstico De Parásitos Gastrointestinales En Caninos Y Felinos: Estudio Retrospectivo En Dos Laboratorios Veterinarios	2021	2005		Costa Rica	57	[68]
RSM-45	Ectoparasitos E Helmintos Intestinais Em Felis Catus Domesticus, Da Cidade De Lages, SC, Brasil E Aspectos Sócioeconômicos E Culturais Das Famílias Dos Proprietários Dos Animais	2009	2005	Lages	Brazil	111	[69]
RSM-46	Endoparasite Prevalence And Infection Risk Factors Among Cats In An Animal Shelter In Estonia	2021	2015	Tartu	Estonia	290	[70]
RSM-47	Endoparasite Prevalence And Recurrence Across Different Age Groups Of Dogs And Cats	2009	1997	Pennsylvania	USA	1566	[71]
RSM-48	Endoparasites Detected In Faecal Samples From Dogs And Cats Referred For Routine Clinical Visit In Sardinia, Italy	2017	2011	Sacer	Italy	343	[72]
RSM-49	Endoparasites In Dogs And Cats Diagnosed At The Veterinary Teaching Hospital (VTH) Of The University Of Prince Edward Island Between 2000 And 2017. A Large-Scale Retrospective Study	2020	2000	Prince Edward Island	Canada	2391	[73]
RSM-50	Endoparasites In Dogs And Cats In Germany 1999–2002	2003	1999	Freiburg	Germany	3167	[74]
RSM-51	Endoparasites In Domestic Cats (Felis Catus) In The Semiarid Region Of Northeast Brazil	2023	2023	Sousa	Brazil	207	[75]
RSM-52	Endoparasites Of Cats From The Tirana Area And The First Report On Aelurostrongylus Abstrusus (Railliet, 1898) In Albania	2011	2008	Tirana	Albania	18	[76]
RSM-53	Endoparasites Of Household And Shelter Cats In The City Of Rio De Janeiro, Brazil	2020	2020	Rio de Janeiro	Brazil	393	[77]
RSM-54	Enteric Parasites Of Free-Roaming, Owned, And Rural Cats In Prairie Regions Of Canada	2015	2015		Canada	219	[78]
RSM-55	ENTEROPARÁSITOS EN PERROS (Canis Familiaris) Y GATOS (Felis Catus) DE LA PROVINCIA DE PUNO	2013	2013	Puno	Peru	96	[79]
RSM-56	Enteroparasitos Encontrados Em Cães E Gatos Atendidos Em Duas Clínicas Veterinárias Nacidade De Manaus, AM	2012	2010	Manaus	Brazil	13	[80]
RSM-57	Epidemiological Survey Of Zoonotic Helminths In Feral Cats I2016n Gran Canaria Island (Macaronesian Archipelago-Spain)	2016	2016	Gran Canaria island	Spain	48	[81]
RSM-58	Epidemiological Survey On Gastrointestinal And Pulmonary Parasites In Cats Around Toulouse (France)	2022	2015		France	498	[82]
RSM-59	Epidemiology Of *Toxocara* Spp. In Stray Dogs And Cats In Dublin, Ireland	1994	1990	Dublin	Ireland	181	[83]
RSM-60	Estudio Coprológico De Parasitosis En Gatos Del Área Periurbana De La Ciudad De Murcia Y Sus Implicaciones Zoonósicas	2017	2017	Murcia	Spain	61	[84]
RSM-61	Estudio De La Prevalencia De Parásitos Gastrointestinales Zoonosicos En Perros Y Gatos En El Barrio Carapungo De La Ciudad De Quito.	2010	2010	Quito	Ecuador	32	[85]
RSM-62	Estudio Retrospectivo De Casos De Parasitosis Gastrointestinales Presentados En Caninos Y Felinos En La Clínica Veterinaria Zooluciones Versátiles En La Ciudad De Bogotá	2021	2010	Bogotá	Colombia	38	[86]
RSM-63	Estudo Das Parasitoses Gastrintestinais De Cães E Gatos Domésticos No Município De São José Dos Campos—Sp.	2005	2004	São José dos Campos	Brazil	20	[87]
RSM-64	Fecal Survey Of Parasites In Free-Roaming Cats In Northcentral Oklahoma, United States	2018	2015	Oklahoma	USA	846	[88]
RSM-65	Feline Gastrointestinal Parasitism In Greece: Emergent Zoonotic Species And Associated Risk Factors	2018	2016	Macedonia, Islands, Central Greece, Epirus, Thrace, Peloponnesus, Thessaly	Greece	1150	[89]
RSM-66	Feline Immunodeficiency Virus, Feline Leukaemia Virus, Toxoplasma Gondii, And Intestinal Parasitic Infections In Taiwanese Cats	1990	1990		Taiwan	95	[90]
RSM-67	Feline Intestinal Parasites In Finland: Prevalence, Risk Factors And Anthelmintic Treatment Practices	2012	2009		Finland	411	[91]
RSM-68	Feline Parasites And The Emergence Of Feline Lungworm In The Portland Metropolitan Area, Oregon, USA 2016–2017	2021	2016	Portland	USA	126	[92]
RSM-69	First Report Of Echinococcus Multilocularis In Cats In Poland: A Monitoring Study In Cats And Dogs From A Rural Area And Animal Shelter In A Highly Endemic Region	2019	2017	Province Podkarpackie	Poland	67	[93]
RSM-70	Freqüência De Helmintos Em Gatos De Uberlândia, Minas Gerais	2004	2000	Uberlândia	Brazil	50	[94]
RSM-71	Frecuencia De Parásitos Gastrointestinales En Animales Domésticos Diagnosticados En Yucatán, México.	2001	1984	Yucatan	Mexico	46	[95]
RSM-72	Frecuencia De Parásitos Gastrointestinales En Felinos Domésticos (Felis Catus) En El Distrito De Jesús María—Lima	2022	2021	Lima	Peru	87	[96]
RSM-73	Frequência De Endoparasitas Em Gatos Internados Em Quatro Clínicas De Cascavel, Paraná	2018	2018	Cascavel	Brazil	23	[97]
RSM-74	Freqüência De Enteroparasitas Em Amostras Fecais De Cães E Gatos Dos Municípios Do Rio De Janeiro E Niterói	2005	1999		Brazil	40	[98]
RSM-75	Frequência De Helmintos Diagnosticados Em Cães E Gatos No Laboratório De Doenças Parasitárias Da Faculdade De Veterinária/Ufpel	2022	2019	Pelotas	Brazil	69	[99]
RSM-76	Frequência De Parasitas Gastrointestinais Em Cães E Gatos Domunicípio De Londrina, PR, Com Enfoque Em Saúde Pública	2013	2000	Londrina	Brazil	378	[100]
RSM-77	Freqüência De Parasitas Intestinais Em Cães (Canis Familiaris) E Gatos (Felis Catus Domestica) Em Araçatuba, São Paulo	1995	1992	Aracatuba	Brazil	32	[101]
RSM-78	Frequência De Parasitos Gastrintestinais, Presentes Em Fezes De Cães E Gatos, Analisadas No Laboratório De Doenças Parasitárias Da Ufpel, Durante O Ano De 2017	2019	2017	Pelotas	Brazil	25	[102]
RSM-79	Frequency Of Gastrointestinal Parasites In Cats Seen At The University Of São Paulo Veterinary Hospital, Brazil	2016	2005	Sao Paulo	Brazil	502	[103]
RSM-80	Freqüência De Parasitos Gastrintestinais Em Cães E Gatos Atendidos Em Hospital-Escola Veterinário Da Cidade De São Paulo	2007	2000	Sao Paulo	Brazil	327	[104]
RSM-81	Gastrointestinal Helminth Parasites In Stray Cats From The Mid-Ebro Valley, Spain	1998	1989	Zaragoza	Spain	58	[105]
RSM-82	Gastrointestinal Helminth Parasites Of Pets: Retrospective Study At The Veterinary Teaching Hospital, IPB University, Bogor, Indonesia	2023	2014	Bogor	Indonesia	171	[106]
RSM-83	Gastrointestinal Helminthes In Stray Cats (Felis Catus) From Aizawl, Mizoram, India	2011	2005	Aizawl	India	27	[107]
RSM-84	Gastrointestinal Helminthic Parasites Of Stray Cats (Felis Catus) In Northwest Iran	2021	2014	Meshkin-Shahr	Iran	104	[108]
RSM-85	Gastrointestinal Helminths And Ectoparasites In The Stray Cats (Felidae: Felis Catus) Of Ahar Municipality, Northwestern Iran	2017	2013	Ahar	Iran	51	[109]
RSM-86	Gastrointestinal Helminths Of Cat (Felis Catus) In Kashmir Valley, India.	2020	2017	Kashmir Valley	India	887	[110]
RSM-87	Gastrointestinal Parasite Infection In Cats In Daegu, Republic Of Korea, And Efficacy Of Treatment Using Topical Emodepside/Praziquantel Formulation	2019	2012	Daegu	Republic of Korea	407	[111]
RSM-88	Gastrointestinal Parasites In Dogs And Cats In Line With The One Health’ Approach	2022	2018	Pernambuco	Brazil	105	[112]
RSM-89	Gastrointestinal Parasites In Feral Cats And Rodents From The Fernando De Noronha Archipelago, Brazil	2017	2016	Fernando de Noronha	Brazil	37	[113]
RSM-90	Gastrointestinal Parasites In Rural Dogs And Cats In Selangor And Pahang States In Peninsular Malaysia	2014	2011		Malaysia	28	[114]
RSM-91	Gastrointestinal Parasites In Shelter Cats Of Central Italy	2019	2011	Latium, Tuscany	Italy	132	[115]
RSM-92	Gastrointestinal Parasites In Stray And Shelter Cats In The Municipality Of Rio De Janeiro, Brazil	2017	2014	Rio de Janeiro	Brazil	263	[116]
RSM-93	Gastrointestinal Parasites Of Cats In Brazil: Frequency And Zoonotic Risk	2016	2016	Pernambuco	Brazil	173	[117]
RSM-94	Gastrointestinal Parasites Of Cats In Denmark Assessed By Necropsyand Concentration Mcmaster Technique	2015	2014		Denmark	99	[118]
RSM-95	Gastrointestinal Parasites Of Dogs And Cats In A Refuge In Nakhon Nayok, Thailand	2014	2014	Nakhon Nayok	Thailand	300	[119]
RSM-96	Gastrointestinal Parasites Of Domestic Cats In Perth, Western Australia	2003	2001	Pert	Australia	418	[120]
RSM-97	Gastrointestinal Parasites Of Feral Cats From Christmas Island	2008	2008	Christmas Island	Australia	28	[121]
RSM-98	Gastro-Intestinal Parasites Of Feral Cats In New South Wales	1976	1969	New South Wales	Australia	146	[122]
RSM-99	Gastrointestinal Parasites Of Stray Cats In Kashan, Iran	2009	2004	Kashan	Iran	113	[123]
RSM-100	Gastrointestinal Parasites Of Cats In Egypt: High Prevalence High Zoonotic Risk	2022	2021	Gharbia	Egypt	143	[124]
RSM-101	Giardia Is The Most Prevalent Parasitic Infection In Dogs And Cats With Diarrhea In The City Of Medellín, Colombia	2019	2018	Medellin	Colombia	203	[125]
RSM-102	Helminth And Protozoan Parasites In Dogs And Cats In Belgium	1991	1980		Belgium	30	[126]
RSM-103	Helminth Burden In Stray Cats From Thessaloniki, Greece	2014	2010	Thessaloniki	Greece	2015	[127]
RSM-104	Helminth Parasites And Arthropods Of Feral Cats	1981	1981		Australia	327	[128]
RSM-105	Helminth Parasites Of Cats From The Vientiane Province, Laos, As Indicators Of The Occurrence Of Causative Agents Of Human Parasitoses	2003	1989		Laos	55	[129]
RSM-106	Helminth Parasites Of Dogs And Cats And Toxoplasmosis Antibodies In Cats In Swansea, South Wales	1978	1977	Swansea	United Kingdom	46	[130]
RSM-107	Helminth Parasites Of The House Cat, Felis Catus, In Connecticut, U.S.A	2003	2003	Connecticut	USA	450	[131]
RSM-108	Helmintofauna De Gatos (Felis Silvestres Catus, Linnaeus, 1758) Da Região Metropolitana De Cuiabá	2012	2010		Brazil	146	[132]
RSM-109	Helmintofauna Parasitária Em Gatos Errantes De Lages, Santa Catarina, Brasil	2021	2012	Lages	Brazil	97	[133]
RSM-110	High Prevalence Of Covert Infection With Gastrointestinal Helminths In Cats	2015	2010	Oklahoma	USA	116	[134]
RSM-111	High Prevalence Of Helminth Parasites In Feral Cats In Majorca Island (Spain)	2009	2008	Majorca Island	Spain	58	[135]
RSM-112	Implications Of Zoonotic And Vector-Borne Parasites To Free-Roaming Cats In Central Spain	2018	2014		Spain	459	[136]
RSM-113	Importation Of Cats And Risk Of Parasite Spread: A Caribbean Perspective And Case Study From St Kitts	2020	2018		Saint Kitts and Nevis	74	[137]
RSM-114	Incidencia De Parásitos Gastrointestinales En Gatos En La Ciudad De Guayaquil	2013	2013	Guayaquil	Ecuador	1200	[138]
RSM-115	Incidencia De Parásitos En Gatos (Felis Silvestris Catus) En El Centro De Bienestar Animal Tecámac Municipio De Tecámac	2020	2020	Tecamac	Mexico	60	[139]
RSM-116	Incidencia De *Toxocara cati* En Felinos Domésticos De La Parroquia Veracruz, Cantón Pastaza, Provincia De Pastaza	2023	2023	Puyo	Ecuador	55	[140]
RSM-117	Infecções Por Parasitos Gastrintestinais Em Gatos Domésticos De Araguaína, Tocantins	2017	2017	Araguaina	Brazil	53	[141]
RSM-118	Infection Status With Helminthes In Feral Cats Purchased From A Market In Busan, Republic Of Korea	2005	1996	Busan	Republic of Korea	438	[142]
RSM-119	Infestação Por Ancilostomídeos E Toxoçarídeos Em Cães E Gatos Apreendidos Em Vias Públicas, São Paulo (Brasil)	1988	1980	Sao Paulo	Brazil	940	[143]
RSM-120	Insights To Helminth Infections In Food And Companion Animals In Bangladesh: Occurrence And Risk Profiling	2022	2020		Bangladesh	10	[144]
RSM-121	Internal Parasites Of Feral Cats From The Tasmanian Midlands And King Island	1976	1973	Midlands	Australia	107	[145]
RSM-122	Intestinal And Lung Parasites In Owned Dogs And Cats From Central Italy	2012	2008	Pisa	Italy	81	[146]
RSM-123	Intestinal Helminthic Infections Of Cats In Calabar, Nigeria	1988	1988	Calabar	Nigeria	52	[147]
RSM-124	Intestinal Helminths Of Cats In The Kainji Lake Area, Nigeria	1986	1986		Nigeria	83	[148]
RSM-125	Intestinal Helminths Of Feral Cat Populations From Urban And Suburban Districts Of Qatar	2010	2006	Doha	Qatar	658	[149]
RSM-126	Intestinal Parasites And Fecal Cortisol Metabolites In Multi-Unowned-Cat Environments: The Impact Of Housing Conditions	2021	2015		Spain	368	[150]
RSM-127	Intestinal Parasites And Lungworms In Stray, Shelter And Privately Owned Cats Of Switzerland	2019	2012		Switzerland	664	[151]
RSM-128	Intestinal Parasites And Risk Factors In Dogs And Cats From Rio De Janeiro, Brazil	2021	2017	Rio de Janeiro	Brazil	208	[152]
RSM-129	Intestinal Parasites In Dogs And Cats From The District Of Évora, Portugal	2011	2007	Evora	Portugal	20	[153]
RSM-130	Parásitos Intestinales En Caninos Y Felinos Con Cuadros Digestivos En Santiago, Chile. Consideraciones En Salud Pública	2006	1996	Santiago de Chile	Chile	230	[154]
RSM-131	Intestinal Parasites Of Cats Purchased In Seoul	1993	1993	Seoul	Republic of Korea	41	[155]
RSM-132	Intestinal Parasites Of Owned Dogs And Cats From Metropolitan And Micropolitan Areas: Prevalence, Zoonotic Risks, And Pet Owner Awareness In Northern Italy	2014	2010		Italy	127	[156]
RSM-133	Estudo De Endoparasitos E Ectoparasitos Em Gatos Domésticos De Área Urbana	2023	2023	Aracatuba	Brazil	61	[157]
RSM-134	Intestinal Parasites Of Pets And Other House-Kept Animals In Moscow	2019	2012	Moscow	Russia	1261	[158]
RSM-135	Intestinal Parasitic Infection In Multi-Cat Shelters In Catalonia	2017	2012	Catalonia	Spain	160	[159]
RSM-136	Investigations On The Endoparasite Fauna Of The Domestic Cat In Eastern Brandenburg [Untersuchungen Zur Endoparasitenfauna Der Hauskatze In Ostbrandenburg]	1997	1993	Brandenburg	Germany	155	[160]
RSM-137	Is There Any Change In The Prevalence Of Intestinal Or Cardiopulmonary Parasite Infections In Companion Animals (Dogs And Cats) In Germany Between 2004–2006 And 2015–2017? An Assessment Of The Impact Of The First ESCCAP Guidelines	2022	2015		Germany	72,200	[161]
RSM-138	Levels Of *Toxocara* Infections In Dogs And Cats From Urban Vietnam Together With Associated Risk Factors For Transmission	2016	2014	Hanoi	Vietnam	253	[162]
RSM-139	Macroparasite Communities In Stray Cat Populations From Urban Cities In Peninsular Malaysia	2013	2007	Kuala Lumpur, Georgetown, Kuantan, Malaca	Malaysia	543	[163]
RSM-140	Molecular Detection Of *Cryptosporidium* Spp. And The Occurrence Of Intestinal Parasites In Fecal Samples Of Naturally Infected Dogs And Cats	2018	2017	Santa Maria	Brazil	49	[164]
RSM-141	Molecular Evaluation Of *Toxocara* Species In Stray Cats Using Loop-Mediated Isothermal Amplification (Lamp) Technique As A Rapid, Sensitive And Simple Screening Assay	2021	2018	Khorramabad	Iran	95	[165]
RSM-142	National Study Of The Gastrointestinal Parasites Of Dogs And Cats In Australia	2008	2004		Australia	1063	[166]
RSM-143	Occurrence And Clinical Significance Of Aelurostrongylus Abstrusus And Other Endoparasites In Danish Cats	2016	2015		Denmark	259	[167]
RSM-144	Occurrence And Zoonotic Potential Of Endoparasites In Cats Of Cyprus And A New Distribution Area For Troglostrongylus Brevior	2017	2017		Republic of Cyprus	185	[168]
RSM-145	Occurrence Of Canine And Feline Extra-Intestinal Nematodes In Key Endemic Regions Of Italy	2019	2015	Abruzzo, Lazio, Molise, Marche, SanPietro Island, Piedmont, Veneto, Friuli-Venezia Giulia	Italy	1000	[169]
RSM-146	Ocorrência De Parasitos Gastrointestinais E Fatores De Risco De Parasitismo Em Gatos Domésticos Urbanos De Santa Maria, RS, Brasil	2013	2011	Santa Maria	Brazil	191	[170]
RSM-147	Seroprevalences Of Antibodies Against Bartonella Henselae And Toxoplasma Gondii And Fecal Shedding Of Cryptosporidium Spp, Giardia Spp, And *Toxocara cati* In Feral And Pet Domestic Cats	2004	2004	North Carolina	USA	153	[171]
RSM-148	Stray Dogs And Cats As Potential Sources Of Soil Contamination With Zoonotic Parasites	2017	2011	Lodz	Poland	68	[17]
RSM-149	Enquête Sur Le Parasitisme Digestif Des Chiens Et Des Chats De Particuliers De La Région Parisienne	2000	1998	Paris	France	34	[172]
RSM-150	Survey Of Helminth Parasites Of Cats From The Metropolitan Area Of Cuiabá, Mato Grosso, Brazil	2013	2010	Cuiaba	Brazil	146	[173]
RSM-151	Ocorrência De Parasitos Gastrintestinais Em Amostras Fecais De Felinos No Município De Andradina, São Paulo	2009	2009	Andradina	Brazil	51	[174]
RSM-152	Ocorrência De Parasitos Gastrintestinais Em Fezes De Gatos Das Cidades De São Paulo E Guarulhos	2002	2002		Brazil	138	[175]
RSM-153	Ocorrência De Protozoários E Helmintos Em Amostras De Fezes De Cães E Gatos Da Cidade De São Paulo	1999	1991	Sao Paulo	Brazil	187	[176]
RSM-154	Occurrence Of Toxoplasma Gondii And Other Gastrointestinal Parasites In Free-Roaming Cats From The Rio De Janeiro Zoo	2023	2023	Rio de Janeiro	Brazil	51	[177]
RSM-155	Ocorrência De Endoparasitas Com Potencial Zoonótico De Transmissão Em Fezes De Gatos (Felis Catus Domesticus Linnaeus, 1758) Domiciliados Na Área Urbana E Região Metropolitana De Castro—Paraná—Brasil	2012	2012	Castro	Brazil	38	[178]
RSM-156	Ocorrência De Endoparasitos Em Gatos De Cuiabá, Mato Grosso, Brasil.	2011	2009	Cuiaba	Brazil	50	[179]
RSM-157	Ocorrência De Nematódeos E Protozoários Em Gatos Com Tutores Da Cidade De Porto Alegre, RS, Brasil.	2020	2018	Porto Alegre	Brazil	266	[180]
RSM-158	Ocorrência De Parasitas Gastrintestinais De Gatos (Felis Catus) Domiciliados Nos Municípios De Patos-PB E Parelhas-RN.	2017	2008	Patos, Parelhas	Brazil	30	[181]
RSM-159	Ocorrência De Parasitas Gastrintestinais Em Fezes De Cães E Gatos, Curitiba-Pr	2005	2005	Curitiba	Brazil	30	[182]
RSM-160	Ocorrência De Parasitos Em Gatos (Felis Catus Domesticu5) E Pombos (Columba Livia) Procedentes De Algumas Localidades De Minas Gerais	1973	1973	Minas Gerais	Brazil	15	[183]
RSM-161	Ocorrência De Parasitas Gastrintestinais Em Cães E Gatos Na Rotina Do Laboratório De Enfermidades Parasitárias Da Fmvz/Unesp-Botucatu, Sp.	2008	2002	Botucatu	Brazil	140	[184]
RSM-162	One-Year Parasitological Screening Of Stray Dogs And Cats In County Dublin, Ireland	2018	2016	Dublin	Ireland	289	[185]
RSM-163	Parasitas Respiratórios, Gastrointestinais E Auriculares Em Gatos De Colónia, Na Casa Dos Animais De Lisboa	2020	2020	Lisbon	Portugal	47	[186]
RSM-164	Parasite Communities In Stray Cat Populations From Lisbon, Portugal	2014	2009	Lisbon	Portugal	120	[187]
RSM-165	Parasite Prevalence In Fecal Samples From Shelter Dogs And Cats Across The Canadian Provinces	2015	2015		Canada	636	[188]
RSM-166	Parasite Prevalence In Free-Ranging Farm Cats, Felis Silvestris Catus	1996	1989		United Kingdom	11	[189]
RSM-167	Parasite Prevalence Survey In Shelter Cats In Citrus County, Florida	2017	2017	Florida	USA	76	[190]
RSM-168	Parasite Richness And Abundance In Insular And Mainland Feral Cats: Insularity Or Density?	2001	1997	Lyon, Kerguelen	France	133	[191]
RSM-169	Parasites And Zoonotic Bacteria In The Feces Of Cats And Dogs From Animal Shelters In Carinthia, Austria	2023	2023	Carinthia	Austria	130	[192]
RSM-170	Parasites Of Domestic Owned Cats In Europe: Co-Infestations And Risk Factors	2014	2012	Budapest	Hungry, Italy, Romania, France, Austria, Spain, Belgium	300	[193]
RSM-171	Parasites Of Feral Cats From Southern Tasmania And Their Potential Significance	1997	1997		Australia	39	[194]
RSM-172	Parasites Of Stray Cats (Felis Domesticus L., 1758) On St. Kitts, West Indies	2010	2005	Basseterre	Saint Kitts and Nevis	100	[195]
RSM-173	Parasitic Infections Of Domestic Cats, Felis Catus, In Western Hungary	2013	2013		Hungry	235	[196]
RSM-174	Exame Parasitológico De Fezes De Gatos (Felis Catus Domesticus) Domiciliados E Errantes Da Região Metropolitana Do Rio De Janeiro, Brasil	2003	2003	Rio de Janeiro	Brazil	131	[197]
RSM-175	Survey Of Infectious And Parasitic Diseases In Stray Cats At The Lisbon Metropolitan Area, Portugal	2010	2003	Lisbon	Portugal	74	[198]
RSM-176	Survey On The Prevalence Of Intestinal Parasites In Domestic Cats (Felis Catus Linnaeus, 1758) In Central Nepal	2023	2020	Ratnanagar	Nepal	90	[199]
RSM-177	Parasitos De Interesse Zoonótico Em Felinos (Felis Catus Domesticus), Campo Grande, Mato Grosso Do Sul	2016	2014	Campo Grande	Brazil	210	[200]
RSM-178	Técnica De Centrífugo-Flutuação Com Sulfato De Zinco No Diagnóstico De Helmintos Gastrintestinais De Gatos Domésticos	2007	2004	Rio de Janeiro	Brazil	13	[201]
RSM-179	The First Study On The Prevalence Of Gastrointestinal Parasites In Owned And Sheltered Cats In Yangon, Myanmar	2023	2022	Yangon	Myanmar	230	[202]
RSM-180	Ocena Zależności Zarażenia Pasożytami Wewnętrznymi Psów I Kotów Od Przygotowania Hodowlano–Weterynaryjnego Wła–Ścicieli	2008	2008	Olsztyn	Poland	35	[203]
RSM-181	PARASITOS GASTRINTESTINAIS EM Felis Catus Linnaeus, 1758 DE MOSSORÓ, RN	2023	2022	Mossoro	Brazil	72	[204]
RSM-182	Parasitos Gastrintestinais Em Fezes De Gatos Domiciliados No Município De Pelotas, RS, Brasil	2021	2018	Pelotas	Brazil	60	[205]
RSM-183	Parasitos Gastrintestinais Em Gatos Da Cidade De Porto Alegre, Rio Grande Do Sul	2017	2014	Porto Alegre	Brazil	339	[206]
RSM-184	Parasitos Gastrointestinais De Caninos E Felinos: Uma Questão De Saúde Pública	2021	2015		Brazil	78	[207]
RSM-185	Parásitos Intestinales En Perros Y Gatos Con Dueño De La Ciudad De Barranquilla, Colombia	2018	2014	Barranquilla	Colombia	45	[208]
RSM-186	Parásitos Zoonóticos Presentes En Gatos Domésticos (Felis Silvestris Catus) En Un Centro De Control Canino Y Felino, En Nuevo León; México	2021	2020	New Leon	Mexico	189	[209]
RSM-187	Parasitoses Gastrointestinais E Pulmonares Em Canídeos E Felídeos Da Região Oeste De Portugal Continental	2017	2017		Portugal	70	[210]
RSM-188	Parasitoses Pulmonares E Gastrointestinais Em Felinos Domésticos No Minho, Portugal	2016	2016	Viana de Castelo, Guimaraes, Ponte de Lima, Vila Verde, Braga, Amares	Portugal	68	[211]
RSM-189	Parasitosis Intestinales En Gatos De Querétaro	2023	2022	Queretaro	Mexico	180	[212]
RSM-190	Parasitosis Zoonóticas En Mascotas Caninas Y Felinas De Niños De Educación Primaria Del Cono Norte De Lima, Perú	2011	2011	Lima	Peru	49	[213]
RSM-191	Presence Of *Toxocara* Eggs On The Hair Of Dogs And Cats	2013	2010	Ankara	Turkey	30	[214]
RSM-192	Presence Of *Toxocara* Spp. In Domestic Cats In The State Of Mexico	2016	2016	Mexico City	Mexico	229	[215]
RSM-193	Prevalence And Associated Risk Factors Of Intestinal Parasites In Rural High-Mountain Communities Of The Valle Del Cauca—Colombia	2020	2020	Valle del Cauca	Colombia	7	[216]
RSM-194	Prevalence And Molecular Characterization Of *Toxocara cati* Infection In Feral Cats In Alexandria City, Northern Egypt	2021	2018	Alexandria	Egypt	100	[217]
RSM-195	Prevalence And Molecular Characterization Of Toxoplasmagondii And *Toxocara cati* Among Stray And Household Cats And Cat Owners In Tehran, Iran	2022	2017	Tehran	Iran	165	[218]
RSM-196	Prevalence And Public Health Relevance Of Enteric Parasites In Domestic Dogs And Cats In The Region Of Madrid (Spain) With An Emphasis On Giardia Duodenalis And Cryptosporidium Sp.	2023	2017	Madrid	Spain	35	[219]
RSM-197	Prevalence And Risk Factors Associated With Cat Parasites In Italy: A Multicenter Study	2021	2019		Italy	987	[220]
RSM-198	Prevalence And Risk Factors Associated With Endoparasitosis Of Dogs And Cats In Espírito Santo, Brazil	2016	2016	Espirito Santo	Brazil	160	[221]
RSM-199	Prevalence And Risk Factors For Patent *Toxocara* Infections In Cats And Cat Owners’ Attitude Towards Deworming	2016	2010		Netherlands	670	[311]
RSM-200	Prevalence And Risk Factors Of Intestinal Parasites In Cats From China	2015	2013	Henan, Beijing	China	360	[222]
RSM-201	Prevalence Of Antibodies To Toxoplasma Gondii And Intestinal Parasites In Stray, Farm And Household Cats In Spain	2004	2004		Spain	382	[223]
RSM-202	Prevalence Of Cryptosporidian Infection In Cats In Turin And Analysis Of Risk Factors	2007	2007	Turin	Italy	200	[224]
RSM-203	Prevalence Of Endoparasites In Household Cat (Felis Catus) Populations From Transylvania (Romania) And Association With Risk Factors	2010	2007	Transylvania	Romania	414	[225]
RSM-204	Prevalence Of Endoparasites In Northern Mississippi Shelter Cats	2019	2017	Mississippi	USA	55	[226]
RSM-205	Prevalence Of Endoparasites In Stray And Fostered Dogs And Cats In Northern Germany	2012	2012	Lower Saxony	Germany	837	[227]
RSM-206	Prevalence Of Endoparasitic And Viral Infections In Client-Owned Cats In Metropolitan Bangkok, Thailand, And The Risk Factors Associated With Feline Hookworm Infections	2021	2014	Bangkok	Thailand	509	[228]
RSM-207	Prevalence Of Enteric Zoonotic Agents In Cats Less Than 1 Year Old In Central New York State	2001	1998	New York	USA	263	[229]
RSM-208	Prevalence Of Enteric Zoonotic Organisms In Cats	2000	1993	Colorado	USA	206	[230]
RSM-209	Prevalence Of Faecal-Borne Parasites In Colony Stray Cats In Northern Italy	2013	2008	Milan	Italy	139	[231]
RSM-210	Prevalence Of Fecal-Borne Parasites Detected By Centrifugal Flotation In Feline Samples From Two Shelters In Upstate New York	2011	2006	New York	USA	1629	[232]
RSM-211	Prevalence Of Fleas And Gastrointestinal Parasites In Free-Roaming Cats In Central Mexico	2013	2010		Mexico	358	[233]
RSM-212	Prevalence Of Gastro-Intestinal And Haemoparasitic Infections Among Domestic Cats Of Kerala	2023	2023	Kerala	India	122	[234]
RSM-213	Prevalence Of Gastrointestinal Helminth Parasites Of Zoonotic Significance In Dogs And Cats In Lower Northern Thailand	2017	2014		Thailand	180	[235]
RSM-214	Prevalencia De Helmintos Gastrointestinales En Gatos Admitidos En La Policlínica Veterinaria De La Universidad Del Zulia	2008	2004	Zulia	Venezuela	64	[236]
RSM-215	The Occurrence Of Endoparasites In Slovakian Household Dogs And Cats	2021	2018	Bratislava	Slovakian	50	[237]
RSM-216	The Parasite Fauna Of Stray Domestic Cats (Felis Catus) In Dubai, United Arab Emirates	2009	2004	Dubai	United Arab Emirates	240	[238]
RSM-217	The Prevalence Of Endoparasites Of Free Ranging Cats (Felis Catus) From Urban Habitats In Southern Poland	2020	2020	Kraków	Poland	81	[239]
RSM-218	Prevalence Of Gastrointestinal Parasites In Domestic Cats (Felis Catus) Diagnosed By Different Coproparasitological Techniques In The Municipality Of Seropédica, Rio De Janeiro	2023	2020	Rio de Janeiro	Brazil	237	[240]
RSM-219	Prevalencia De Parásitos Gastrointestinales En Gatos Domésticos (Felis Silvestris Catus Schreber, 1775) En La Habana, Cuba	2020	2014	La Havana	Cuba	356	[241]
RSM-220	Prevalence Of Helminth And Coccidian Parasites In Swedish Outdoor Cats And The First Report Of Aelurostrongylus Abstrusus In Sweden: A Coprological Investigation	2017	2017		Sweden	205	[242]
RSM-221	Prevalence Of Hookworm Infection And Strongyloidiasis In Cats And Potential Risk Factor Of Human Diseases	2018	2016	Thasala	Thailand	15	[243]
RSM-222	Prevalence Of Internal Helminthes In Stray Cats (Felis Catus) In Mosul City, Mosul-Iraq	2012	2008	Mosul	Iraq	55	[244]
RSM-223	Prevalence Of Intestinal Endoparasites With Zoonotic Potential In Domestic Cats From Botucatu, SP, Brazil	2017	2011	Botucatu	Brazil	1725	[245]
RSM-224	Prevalence Of Intestinal Nematodes Of Dogs And Cats In The Netherlands	1997	1993	Utrecht, Amersfoort	Netherlands	292	[246]
RSM-225	Prevalence Of Intestinal Parasites Detected In Routine Coproscopic Methods In Dogs And Cats From The Masovian Voivodeship In 2012–2015	2019	2012	Warsaw	Poland	5809	[247]
RSM-226	Prevalence Of Intestinal Parasites In Breeding Cattery Cats In Japan	2016	2013		Japan	342	[248]
RSM-227	Kedilerde Bağırsak Parazitlerinin Yaygınlığı Ve Halk Sağlığı Bakımından Önemi	2016	2015	Kırıkkale	Turkey	100	[249]
RSM-228	Prevalence Of Intestinal Parasites In Companion Animals In Ontario And Quebec, Canada, During The Winter Months	2008	2008		Canada	47	[250]
RSM-229	Prevalence Of Intestinal Parasites In Dogs And Cats From The Kvarner Region In Croatia	2023	2019	Kvarner	Croatia	64	[251]
RSM-230	Prevalence Of Intestinal Parasites In Dogs And Cats Under Veterinary Care In Porto Alegre, Rio Grande Do Sul, Brazil	2007	2002	Porto Alegre	Brazil	288	[252]
RSM-231	Prevalence Of Intestinal Parasites In Feral Cats In Some Urban Areas Of England	1981	1978		United Kingdom	92	[253]
RSM-232	Prevalence Of Intestinal Parasites In Pet Shop Kittens In Japan	2013	2011		Japan	555	[254]
RSM-233	Prevalence Of Intestinal Parasites In Private-Household Cats In Japan	2012	2008		Japan	942	[255]
RSM-234	Prevalence Of Intestinal Parasites In Shelter Dogs And Cats In The Metropolitan Area Of Barcelona (Spain)	2009	1999	Barcelona	Spain	50	[256]
RSM-235	Prevalence Of Intestinal Parasites, Risk Factors And Zoonotic Aspects In Dog And Cat Populations From Goiás, Brazil	2023	2020	Goiás	Brazil	55	[257]
RSM-236	Prevalence Of Major Digestive And Respiratory Helminths In Dogs And Cats In France: Results Of A Multicenter Study	2022	2017		France	425	[3]
RSM-237	Prevalence Of Protozoa And Gastrointestinal Helminthes In Stray Cats In Zanjan Province, North-West Of Iran	2009	2007	Zanjan	Iran	100	[258]
RSM-238	Prevalence Of Selected Bacterial And Parasitic Agents In Feces From Diarrheic And Healthy Control Cats From Northern California	2012	2007	California	USA	269	[259]
RSM-239	Prevalence Of Selected Zoonotic And Vector-Borne Agents In Dogs And Cats In Costa Rica	2011	2009	San Isidro de El General	Costa Rica	9	[260]
RSM-240	Prevalence Of Some Gastrointestinal Parasites In Cats In The Perth Area	1983	1978	Perth	Australia	752	[261]
RSM-241	Prevalence Of Species Of *Toxocara* In Dogs, Cats And Red Foxes From The Poznan Region, Poland	2001	1997	Poznan	Poland	105	[262]
RSM-242	Prevalence Of *Toxocara cati* And Other Intestinal Helminths In Stray Cats In Shiraz, Iran	2007	2005	Shiraz	Iran	114	[263]
RSM-243	Prevalence Of *Toxocara cati* In Pet Cats And Its Zoonotic Importance In Tabriz City, Iran	2020	2014	Tabriz	Iran	50	[264]
RSM-244	Prevalence Of *Toxocara* Infection In Domestic Dogs And Cats In Urban Environment	2018	2011		Russia	1146	[265]
RSM-245	Prevalence Of *Toxocara* Spp. And Other Parasites In Dogs And Cats In Halifax, Nova Scotia	1978	1971	Halifax	Canada	299	[266]
RSM-246	Prevalence Of Toxocariasis And Its Related Risk Factors In Humans, Dogs And Cats In Northeastern Iran: A Population-Based Study	2019	2017	Khorasan Razavi	Iran	236	[267]
RSM-247	Prevalence Of Toxoplasma Gondii And Other Gastrointestinal Parasites In Domestic Cats From Households In Thika Region, Kenya	2017	2015	Thika	Kenya	103	[268]
RSM-248	Prevalence Of Toxoplasma Gondii And Other Intestinal Parasites In Cats In Tokachi Sub-Prefecture, Japan	2018	2013	Tokachi	Japan	351	[269]
RSM-249	Prevalence Of Toxoplasma Gondii Antibodies And Intestinal Parasites In Stray Cats From Nigde, Turkey	2008	2003	Nigde	Turkey	72	[270]
RSM-250	Prevalence Of Zoonotic Parasites In Feral Cats Of Central Virginia, USA	2018	2016	Virginia	USA	192	[271]
RSM-251	Prevalence Survey Of Gastrointestinal And Respiratory Parasites Of Shelter Cats In Northeastern Georgia, USA	2019	2019	Georgia	USA	103	[272]
RSM-252	Prevalence, Co-Infection And Seasonality Of Fecal Enteropathogens From Diarrheic Cats In The Republic Of Korea (2016–2019): A Retrospective Study	2021	2016		Republic of Korea	2789	[273]
RSM-253	Prevalência De Helmintos Em Gatos (Felis Catus Domesticus) De Goiânia	1974	1973	Goiânia	Brazil	37	[274]
RSM-254	Prevalencia De Infección Por *Toxocara cati* Y Giardia Duodenalis En Gato Domestico	2018	2018	Lima	Peru	70	[275]
RSM-255	Prevalência De Parasitas Gastrointestinais E Cardiorrespiratórios Em Gatos Domésticos Na Área Metropolitana De Lisboa	2020	2020	Lisbon	Portugal	77	[276]
RSM-256	Prevalência De Parasitas Gastrointestinais Em Felinos No Concelho De Vila Nova De Gaia	2022	2021	Vila Nova de Gaia	Portugal	102	[277]
RSM-257	Prevalencia De Parásitos Gastrointestinales En Gatos Domésticos (Felis Catus) En La Parroquia La Matriz Del Cantón Latacunga.	2019	2019	Latacunga	Ecuador	100	[278]
RSM-258	Prevalencia De Parásitos Gastrointestinales En Muestras Coprológicas De Caninos Y Felinos Remitidas Al Laboratorio Ejelab, Risaralda. Abril 2017- Abril 2018	2018	2017	Risaralda	Colombia	85	[279]
RSM-259	Prevalência De Parasitos Intestinais Em Gatos Errantes Em Goiânia—Goiás: Ênfase No Diagnóstico De Toxoplasma Gondii Eavaliação Da Acurácia De Técnicas Parasitológicas	2015	2012	Goiânia	Brazil	154	[280]
RSM-260	Prevalencia De Parásitos Intestinales En Los Habitantes Y Sus Mascotas En Los Barrios Hospital, San Lorenzo, Amanecer Y San Antonio Del Municipio De Amatitlán	2014	2014	Amatitlán	Guatemala	15	[281]
RSM-261	Prevalencia De *Toxocara cati* En Felinos Domésticos (Felis Catus) En El Sector La Venecia Ii Del Distrito Metropolitano De Quito	2021	2021	Quito	Ecuador	100	[282]
RSM-262	Prevalencia De *Toxocara cati* En Gatos Domésticos En El Sector De Balerio Estacio, De La Ciudad De Guayaquil	2018	2018	Guayaquil	Ecuador	80	[283]
RSM-263	Principal Endoparasitoses Of Domestic Cats In Sardinia	2004	2000	Sardinia	Italy	183	[284]
RSM-264	Rastreio De Parasitas Gastrointestinais E Pulmonares Em Canídeos E Felídeos Da Região Autónoma Dos Açores—Ilhas De São Miguel E Terceira	2020	2019	Terceira, São Miguel	Portugal	115	[285]
RSM-265	Rastreio De Parasitas Gastrointestinais E Pulmonares Em Gatos De Gatis Nos Distritos De Lisboa E Setúbal, Portugal	2017	2015	Lisbon	Portugal	260	[286]
RSM-266	Recent Investigation On The Prevalence Of Gastrointestinal Nematodes In Cats From France And Germany	2003	1998		Germany	441	[287]
RSM-267	Results Of Parasitological Examinations Of Faecal Samples From Cats And Dogs In Germany Between 2003 And 2010	2011	2003	Freiburg	Germany	8560	[288]
RSM-268	Retrospective Survey Of Parasitism Identified In Feces Of Client-Owned Cats In North America From 2007 Through 2018	2020	2007		USA	2568	[289]
RSM-269	Risk Factors Associated With Cat Parasites In A Feline Medical Center	2021	2021	Mexico City	Mexico	528	[290]
RSM-270	Role Of Small Mammals In The Epidemiology Of Toxocariasis	1995	1995		Slovakian	116	[291]
RSM-271	The Prevalence Of Giardia And Other Intestinal Parasites In Children, Dogs And Cats From Aboriginal Communities In The Kimberley	1993	1993	Kimberley	Australia	33	[292]
RSM-272	The Prevalence Of Intestinal Helminths In Stray Cats In Central Scotland	1980	1980	Glasgow	United Kingdom	72	[293]
RSM-273	The Prevalence Of Intestinal Nematodes In Cats And Dogs From Lancashire, North-West England	2016	2016	Lancashire	United Kingdom	131	[294]
RSM-274	The Prevalence Of Intestinal Parasites In Dogs And Cats In Calgary, Alberta	2011	2008	Calgary	Canada	153	[295]
RSM-275	The Prevalence Of Intestinal Parasites Of Domestic Cats And Dogs In Vladivostok, Russia During 2014–2017	2018	2014	Vladivostok	Russia	135	[296]
RSM-276	The Prevalence Of Potentially Zoonotic Intestinal Parasites In Dogs And Cats In Moscow, Russia	2023	2018	Moscow	Russia	1350	[297]
RSM-277	The Prevalence Of *Toxocara cati* In Domestic Cats In Mexico City	2003	2003	Mexico City	Mexico	520	[298]
RSM-278	The Prevalence Of *Trichuris* Spp. Infection In Indoor And Outdoor Cats On St. Kitts	2015	2015		Saint Kitts and Nevis	41	[299]
RSM-279	*Toxocara* Canis And *Toxocara cati* In Stray Dogs And Cats In Bangkok, Thailand: Molecular Prevalence And Risk Factors	2022	2022	Bangkok	Thailand	500	[300]
RSM-280	*Toxocara cati* And Other Parasitic Enteropathogens: More Commonly Found In Owned Cats With Gastrointestinal Signs Than In Clinically Healthy Ones	2021	2021	Cluj-Napoca	Romania	137	[301]
RSM-281	*Toxocara cati* Infections In Stray Cats In Northern Iran	2007	2004	Mazandaran	Iran	100	[302]
RSM-282	*Toxocara* Infection In Dogs And Cats In Isfahan Province Of Iran In 2021	2023	2023	Isfahan	Iran	230	[303]
RSM-283	*Toxocara* Nematodes In Stray Cats From Shiraz, Southern Iran: Intensity Of Infection And Molecular Identification Of The Isolates	2013	2011	Shiraz	Iran	30	[304]
RSM-284	What Is The Role Of Swiss Domestic Cats In Environmental Contamination With Echinococcus Multilocularis Eggs?	2023	2022		Switzerland	146	[305]
RSM-285	*Cryptosporidium* Spp. In Dogs And Cats In Poland	2021	2016		Poland	101	[306]
RSM-286	Zoonotic And Other Gastrointestinal Parasites In Cats In Lumajang, East Java, Indonesia	2020	2018		Indonesia	120	[307]
RSM-287	Zoonotic Helminths Parasites In The Digestive Tract Of Feral Dogs And Cats In Guangxi, China	2015	2012	Guangxi	China	39	[308]
RSM-288	Zoonotic Parasites In Fecal Samples And Fur From Dogs And Cats In The Netherlands	2009	2007		Netherlands	63	[309]
RSM-289	Animal Toxocariasis In A Megalopolis Epidemic Aspects	2015	2015		Russia	44	[310]

## Data Availability

Available upon reasonable request.

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
