# Peer review of "Toxocara cati Infection in Cats (Felis catus): A Systematic Review and Meta-Analysis"

_animals, 2024, doi:10.3390/ani14071022_

Round 1

Reviewer 1 Report

Comments and Suggestions for Authors

The manuscript is a systematic review with meta-analysis about Toxocara cati infections in cats and their global prevalence, that also includes evaluations regarding the detection methods and any drawbacks emerging from the analysis method chosen for diagnosis. The approach for references research and selection was comprehensive, adequately planned and comprehensively described. The conclusions emerging from the articles collected and analyzed are interesting, but I would suggest improving the formulation of sentences, especially in Discussion paragraph.

Comments

Lines 270-274

Can you please resentence this section, in order to better clarify your meaning?

Lines 285-292

The authors suggest that the higher prevalence of toxocariasis may be related to socio-economic and cultural factors (lower, middle-income economies, Asian and middle eastern countries. From this point of view, how do they frame the position of the United Kingdom, New Zealand and Denmark highlighted by the bar graph in figure 4? To the best of your knowledge or to what can be found in the literature, is there any other factor in common between the countries in the top positions? Do you think that the number of articles or the number of samples analysed per study for each Country may affect the outcome of this analysis? Are the prevalences reported for each Country calculated on the basis of comparable total numbers of samples analysed? 

Lines 308-315

Are you stating that PCR methods are less sensitive compared to other methods or that more research is needed in order to develop standardised protocols? Again, do you think that the number of studies based on molecular analyses and the total number of samples may affect the reliability of the calculated lower prevalence? Could the lower prevalence recorded with molecular methods be due to a reduced diffusion of these methods, as they are more recently developed and less applied than traditional methods?

Conclusions

Is the significance of toxocariasis solely limited to the risk for humans to get infected?

I would suggest to also underline other aspects of relevance of the infection, such as in the field of animal health and if there is any possible economic implication (e.g. for meat farms).

Author Response

Reviewer 1

The manuscript is a systematic review with meta-analysis about Toxocara cati infections in cats and their global prevalence, that also includes evaluations regarding the detection methods and any drawbacks emerging from the analysis method chosen for diagnosis. The approach for references research and selection was comprehensive, adequately planned and comprehensively described. The conclusions emerging from the articles collected and analyzed are interesting, but I would suggest improving the formulation of sentences, especially in Discussion paragraph.

Thanks for your comments

Comments

Lines 270-274

Can you please resentence this section, in order to better clarify your meaning?

Thanks. We have now corrected and clarified the sentences.

Lines 285-292

The authors suggest that the higher prevalence of toxocariasis may be related to socio-economic and cultural factors (lower, middle-income economies, Asian and middle eastern countries. From this point of view, how do they frame the position of the United Kingdom, New Zealand and Denmark highlighted by the bar graph in figure 4? To the best of your knowledge or to what can be found in the literature, is there any other factor in common between the countries in the top positions? Do you think that the number of articles or the number of samples analysed per study for each Country may affect the outcome of this analysis? Are the prevalences reported for each Country calculated on the basis of comparable total numbers of samples analysed?

We agree on that. And then, we have add this additional text as limitation of the analysis:

Although that, the number of articles and the number of samples analysed per study for some countries would be insufficient to understand the relationships between prevalence and associated factors, despite the fact that the prevalence is weighted in the meta-analysis by number of studies and sample size.

Lines 308-315

Are you stating that PCR methods are less sensitive compared to other methods or that more research is needed in order to develop standardised protocols? Again, do you think that the number of studies based on molecular analyses and the total number of samples may affect the reliability of the calculated lower prevalence? Could the lower prevalence recorded with molecular methods be due to a reduced diffusion of these methods, as they are more recently developed and less applied than traditional methods?

We agree also on that. And for that we have also included a new additional text explaining better this limitation.

Again, although that, the number of articles and the number of samples analysed per study by molecular methods, such as PCR, would be insufficient to understand the differences in sensitivity and specificity of methods, despite the fact that the prevalence is weighted in the meta-analysis by number of studies and sample size. To understand the sensitivity of PCR, specific studies of diagnostic test comparison should be performed, which was clearly out of the objectives of this systematic review, focused on prevalence of T. cati in cats.

Conclusions

Is the significance of toxocariasis solely limited to the risk for humans to get infected?

I would suggest to also underline other aspects of relevance of the infection, such as in the field of animal health and if there is any possible economic implication (e.g. for meat farms).

No. As is clearly stated in the article, toxocariasis may affect multiple species. We agree to expand regarding such implications. Now, included in Conclusions.

Reviewer 2 Report

Comments and Suggestions for Authors

Dear Authors,

The systematic review "Toxocara cati infection in cats (Felis catus): A systematic review and meta-analysis"provides interesting information about the prevalence of Toxocara cati in cats worldwide. I read the manuscript with great interest. The manuscript is well-structured.

It could be accepted after some minor revisions. Study limitations should be added. The authors did not make any reference to the age of cats. This aspect could also be important in the prevalence of T. cati. Age could also be a risk factor for Toxocara cati infection.

Please use in text T. cati instead Toxocara cati.

L 83: T. cati instead T. catis.

Thank you!

Author Response

Reviewer 2

Dear Authors,

The systematic review "Toxocara cati infection in cats (Felis catus): A systematic review and meta-analysis"provides interesting information about the prevalence of Toxocara cati in cats worldwide. I read the manuscript with great interest. The manuscript is well-structured.

Thanks a lot.

It could be accepted after some minor revisions. Study limitations should be added. The authors did not make any reference to the age of cats. This aspect could also be important in the prevalence of T. cati. Age could also be a risk factor for Toxocara cati infection.

Thanks. We agree on that. Then, at the end of discussion we added a paragraph explaining such limitations.

This systematic review has certain limitations, including the fact that we were una-ble of assess the age or gender of cats, as this was not reported in most of the studies. This aspect could also be important in the prevalence and risk of T. cati infection, as has been suggested in T. canis [27].

Please use in text T. cati instead Toxocara cati.

L 83: T. cati instead T. catis.

Done. Corrected.

Thank you!

Reviewer 3 Report

Comments and Suggestions for Authors

The topic of the paper is very interesting, but some improvement would make this "great piece of work" more valuable.

Some comments:

Introduction - (18-12): Toxocariasis - it starts with T. canis, but topic is "T. cati" . /By the way - shouldn't be T. mystax (s. cati)? - minor problem /

20-21 - It seems to me, that T. canis is analyzed here, but not T. cati. One could assume that T. cati is more likely problem of the home garden, etc.

Conclusions (31-35) - some differences between this "version" and conclusions at the end of the text (341-354). Detailes of the life cycle / pathology (...344-347) could be described in "discussion". Conclusions should be very clear.

Lane 50-54: sequence of symptoms must be changed. It would be better to start with lighter to extended ones. What is now: cachexia, emaciation..... but diarrhoea almost at the end.

General question: and what about Toxascaris leonina? Never mentioned enywhere.

Author Response

Reviewer 3

The topic of the paper is very interesting, but some improvement would make this "great piece of work" more valuable.

Thanks a lot.

Some comments:

Introduction - (18-12): Toxocariasis - it starts with T. canis, but topic is "T. cati" . /By the way - shouldn't be T. mystax (s. cati)? - minor problem /

Noticed. Indeed, but toxocariasis is caused by both, T. canis and T. cati.

20-21 - It seems to me, that T. canis is analyzed here, but not T. cati. One could assume that T. cati is more likely problem of the home garden, etc.

  1. cati in cats is the focus of the systematic review. We assessed its prevalence in cats.

Conclusions (31-35) - some differences between this "version" and conclusions at the end of the text (341-354). Detailes of the life cycle / pathology (...344-347) could be described in "discussion". Conclusions should be very clear.

We have revised Discussion and Conclusions now.

Lane 50-54: sequence of symptoms must be changed. It would be better to start with lighter to extended ones. What is now: cachexia, emaciation..... but diarrhoea almost at the end.

Ok. We changed the order.

General question: and what about Toxascaris leonina? Never mentioned enywhere.

The objective of this systematic review was to assess the prevalence of Toxocara cati in cats. No other species in the Toxocara genus, nor in the family Toxocaridae. Nevertheless, we take the opportunity to include a comment in the Discussion about it.
